# A Cross-Sectional Survey on the Association between Dental Health Conditions and University Personality Inventory Scores among University Students: A Single-Center Study in Japan

**DOI:** 10.3390/ijerph19084622

**Published:** 2022-04-12

**Authors:** Shigeo Ishikawa, Naohiko Makino, Hitoshi Togashi, Nanami Ito, Atsushi Tsuya, Makiko Hayasaka, Tsuneo Konta, Naoki Okuyama, Kazuyuki Yusa, Mitsuyoshi Iino

**Affiliations:** 1Department of Dentistry, Oral and Maxillofacial Plastic and Reconstructive Surgery, Faculty of Medicine, Yamagata University, 2-2-2 Iida-nishi, Yamagata 990-9585, Japan; nammzaemon@gmail.com (N.O.); k-yusa@med.id.yamagata-u.ac.jp (K.Y.); m-iino@med.id.yamagata-u.ac.jp (M.I.); 2Yamagata University Health Administration Center, 1-4-12 Kojirakawa-machi, Yamagata 990-8560, Japan; namakino@med.id.yamagata-u.ac.jp (N.M.); htogashi@med.id.yamagata-u.ac.jp (H.T.); nito@cc.yamagata-u.ac.jp (N.I.); mkoashino@jm.kj.yamagata-u.ac.jp (M.H.); 3Institute for Promotion of Medical Science Research, Faculty of Medicine, Yamagata University, 2-2-2 Iida-nishi, Yamagata 990-9585, Japan; tsuya@nike.is.dream.jp; 4Department of Public Health and Hygiene, Graduate School of Medicine, Yamagata University, 2-2-2 Iida-nishi, Yamagata 990-9585, Japan; kkonta@med.id.yamagata-u.ac.jp

**Keywords:** surveys and questionnaires, oral health, temporomandibular joint disorders, mental health, students

## Abstract

This study examined the association between dental health conditions and scores on the University Personality Inventory (UPI) among university students in Japan. Participants were freshmen at Yamagata University between 2010 and 2019. Dental check-ups, including dental caries, periodontal disease, malocclusion, and temporomandibular disorders (TMD), and mental health screening using the UPI were performed; 12,433 students were included in the final analysis. A logistic regression analysis was performed to confirm the association between dental health conditions and >30 UPI scores, which indicate the need to consult mental health professionals. Overall, students who required treatment for TMD had a 3.165-fold higher risk of >30 UPI scores (OR = 3.165, 95% CI = 1.710–5.857). Periodontal disease and TMD in male participants (periodontal disease: OR = 1.329, 95% CI = 1.108–1.595; TMD: OR = 3.014, 95% CI = 1.107–8.204) and TMD in female participants (OR = 2.938, 95% CI = 1.340–6.445) were significant risk factors for >30 UPI scores. Students requiring treatment for TMD were at risk of obtaining >30 UPI scores. Although our study has several limitations, students with subjective symptoms (e.g., disturbance in opening the mouth) should take the UPI test or in some cases consult mental health professionals.

## 1. Introduction

In general, an individual’s university life is the period during which they develop into an adult. Research has shown that university students encounter some specific stressors, such as promotion to the next grade or failure, job placement, and socializing with friends and partners [1,2]. Furthermore, many students begin living on their own when they enter university life [3]; thus, the effect of loneliness on stress also requires attention. Moreover, in the current situation due to the COVID-19 pandemic, university students’ activity has been restrained, and an increasing number of reports have shown that students’ mental health is at risk [4,5,6]. However, due to the pandemic, many students prefer to avoid face-to-face health examinations [4]. Compared with other countries, telemedicine is still uncommon in Japan because of the health insurance system [7,8,9]. Therefore, the risk of delay in the initial management of university students’ mental health has increased. Therefore, it is important to pay attention to university students who may be dealing with stress, and it is also crucial to validate mental health using simplified questionnaires, such as the University Personality Inventory (UPI) discussed below.

The UPI has been used for over half a century in Japan for convenient mental health screening for university students [3,10]. The UPI is a 60-item self-report questionnaire that uses a binary response scale (yes or no). A “yes” response is scored as 1, and a “no” is scored as 0 [3]. Students who obtain a total UPI score above 30 are advised to consult mental health professionals. The validity of the UPI has already been assessed, and it has been widely used in Japan in epidemiological studies regarding university students’ mental health [3,11,12]. However, it has also been pointed out that the UPI is considered complex and lengthy [3]. Some students may find it frustrating to answer 60 questions and may not be keen to complete the entire questionnaire. Thus, it is necessary to employ a simpler tool for mental health screening among university students that can be used as a pre-screening test before the UPI. 

Therefore, we hypothesized the following: students requiring treatment for dental health will have a higher risk of >30 UPI scores, which means they will be advised to consult mental health professionals, compared to those who did not require treatment. If a UPI score above 30 is related to subjective dental symptoms, students with such symptoms may be considered at risk of poor mental health. Although associations between dental symptoms and mental health have been previously reported [13,14,15,16,17], few studies have been conducted in this regard on university students [18,19]. Detecting subjective dental symptoms, such as toothache, gingival bleeding, or disturbance in opening the mouth, can help identify students who require mental health screening.

Therefore, we conducted a cross-sectional study to determine the associations between dental health conditions and UPI scores for freshmen at Yamagata University. To the best of our knowledge, no study has previously examined this association in university students. 

## 2. Materials and Methods

### 2.1. Study Design and Participants

This study was approved by the ethics committee of the Yamagata University School of Medicine (No-2021-185). The participants were freshmen who entered Yamagata University from 2010 to 2019. At Yamagata University, medical and dental check-ups are performed for all freshmen every year. Furthermore, mental health screening using the UPI is also performed for all freshmen every year; students with scores greater than 30 are recommended to consult mental health professionals. We used these data on dental examination and UPI for the present study. Medical and dental check-ups were performed by doctors and dentists, respectively, for each student one month after enrollment. Mental health screening using the UPI was also performed at approximately the same time. The UPI was completed by 18,873 students from 2010 to 2019, and dental check-ups were performed for 14,524 students from 2010 to 2019, except for 2015 and 2016 (the dental examination data for these two years could not be used because of system failure). Overall, 12,433 students who completed both tasks and answered all questions on the UPI were included in the final statistical analysis. 

### 2.2. Measurements

Dental examinations were performed to screen for dental caries, periodontal disease, malocclusion, and temporomandibular disorders (TMD). To minimize the measurement bias, before the dental examinations, dentists met to confirm the diagnostic criteria for dental caries, periodontal disease, malocclusion, and TMD. To detect dental caries and periodontal disease, dentists used a portable mobile dental examination light, dental mirror, dental explorer, and dental probe. Dentists detected malocclusion and temporomandibular disorders by visual and tactile examination. The diagnosis was categorized into three groups: no abnormality, requires observation, and requires treatment. The concrete diagnostic criteria were as follows. Dental caries: no abnormality (no caries), requires observation (subsurface decalcification and non-substance defect), and requires treatment (substance defect). Periodontal disease: no abnormality (healthy periodontal tissue), requires observation (mild periodontal inflammation), and requires treatment (moderate or severe periodontal inflammation). Malocclusion: no abnormality (healthy occlusion), requires observation (mild malocclusion), and requires treatment (moderate or severe malocclusion). TMD: no abnormality (healthy condition), requires observation (temporomandibular joint sound without pain or mild trismus), and requires treatment (TMD-related pain or marked trismus). The UPI comprises 60 self-report items with two response options: yes and no. A response of “yes” is given 1 point, while a “no” is given 0 points. The 60 items include four lie scale items; therefore, a total of 56 items, excluding the lie scales, are analyzed to determine the total score. 

### 2.3. Statistical Analyses

The Kolmogorov–Smirnov test was performed to confirm the distribution. The distribution of participant characteristics was analyzed using the chi-squared test for qualitative variables. Crude odds ratios (ORs) for the risk of a UPI score above 30 were calculated using univariate logistic regression. To examine the independent association between a UPI score above 30 and several parameters, we performed a multivariate logistic regression analysis to estimate the adjusted ORs and 95% confidence intervals (95% CIs). We selected the representative variables that were significant in the univariate analysis (*p* < 0.05). The logistic regression requires there to be little or no multicollinearity among the independent variables. Therefore, we confirmed that the values of variance inflation factors of all independent variables were less than 10. Furthermore, the interaction between the variables in the multivariate logistic regression analysis was tested using a two-way analysis of variance. Statistical significance was set at *p* < 0.05. Statistical analyses were performed using SPSS version 25.0 (IBM Corp., Armonk, NY, USA).

## 3. Results

The *p*-value of the Kolmogorov–Smirnov test was <0.001; therefore, the distribution of data from the UPI was not normal. Table 1 shows the characteristics of the participants in the two groups (those with <30 or >30 UPI scores). The chi-squared test revealed significant differences between the two groups in terms of sex and a diagnosis of TMD. Table 2 shows the characteristics of dental health status stratified according to sex. Although the statistical significance was confirmed from the presence of “Not available” on sex, the distribution of dental health status between “Male” and “Female” was very similar. As shown in Table 3, participants who required treatment for TMD had a 3.165 times higher risk of obtaining a UPI score above 30 compared with those who had no abnormality or required observation (OR = 3.165, 95% CI = 1.710–5.857). In male students, periodontal disease and TMD were significant risk factors for a UPI score above 30. Male students who required treatment for periodontal disease showed a 1.329 times higher risk of obtaining a UPI score above 30 than those who had no abnormality or required observation (OR = 1.329, 95% CI = 1.108–1.595). Furthermore, male participants who required treatment for TMD also had a 3.014 times higher risk of obtaining >30 UPI scores than those who had no abnormality or required observation (OR = 3.014, 95% CI = 1.107–8.204). The interaction effect between “Periodontal disease” and “Temporomandibular disorders” in male participants was not confirmed (*p* = 0.723). In female students, TMD were a significant risk factor for a UPI score above 30; those who required treatment for TMD had a 2.938 times higher risk of obtaining >30 UPI scores than those who had no abnormality or required observation (OR = 2.938, 95% CI = 1.340–6.445).

## 4. Discussion

The present study examined the association between UPI scores and dental health conditions in university students in Japan and revealed that students who required treatment for TMD were at risk for obtaining >30 UPI scores. Furthermore, students who required treatment for TMD displayed subjective symptoms, such as disturbance in opening the mouth, pain in opening the jaw, and tenderness on self-palpation. These symptoms are easy to recognize; therefore, students who can be identified as having subjective symptoms are at risk of having mental health issues and are strongly recommended to take the UPI test or in some cases consult a mental health professional. Considering these points, the results of the present study are noteworthy.

Correlations between dental health conditions and mental health have been reported in several studies [13,14,15,16,17]. In particular, the association between TMD and mental health has been widely reported [20,21,22], although the precise mechanisms between mental health and TMD remain unclear [21]. The etiology of TMD is multifactorial and is well-known to comprise not only biological factors, such as derangements in the temporomandibular joint, but also psychological factors, such as depression, anxiety, and stress [21]. There is a clear link between stress and bruxism, which may explain TMD. Stress has the potential to alter the threshold of pain perception in the central nervous system and increases the intensity of parafunctional habits such as bruxism [21]. Colonna et al. found that almost 50% of 506 participants with poor mental health due to the COVID-19 pandemic reported an increase in bruxism behaviors, and approximately 30% of the participants reported an increase in their TMD symptoms [23]. Carter also reported an association between stress and bruxism among veterans with Gulf War illness [24]; the participants, who had higher levels of perceived stress than the general population, demonstrated a high frequency of teeth grinding and clenching. Bruxism puts pressure on the orofacial muscle, and persistent pressure on this muscle leads to pain in the masseter and temporal muscles. Furthermore, persistent pressure on the mandible also leads to temporomandibular joint pain. These pains are some of the typical symptoms of TMD. Considering these past reports, the results of the present study regarding the association between TMD and the UPI can be considered reasonable and logical.

In this study, periodontal disease was also a significant risk factor for obtaining a UPI score above 30, although this was found only in male participants. Correlations between mental disorders and periodontal disease have also been reported in previous studies [25,26]. It has been shown that an inadequate oral hygiene routine and lack of access to dental services due to poor mental health leads to poor oral hygiene [26]. Furthermore, poor oral hygiene leads to inflammation of gingival tissue and gingival bleeding, also known as periodontal disease. It is unclear why periodontal disease was a significant risk factor for >30 UPI scores in male students but not in female students. Further studies are required to confirm and clarify this. However, considering that the subjective symptoms of periodontal disease, such as gingival pain or bleeding, are very easy to recognize, these symptoms can serve as an important pre-screening test for mental health.

Furthermore, in the present study, malocclusion and dental caries were not significant risk factors for >30 UPI scores. The viewpoint that poor dental health conditions may derive from mental illness does not apply to malocclusion because it is not reasonable to state that malocclusion occurs due to mental illness. However, malocclusion, including jaw deformity, is known to affect mental health [27,28]. A positive correlation has also been reported between dental caries and poor mental health [14]. In addition to periodontal disease, inadequate oral hygiene and lack of access to dental services due to poor mental health may lead to dental caries. At present, it is difficult to identify or discuss the reasons for the results obtained in this study; further research is required to obtain clarification.

This study has several limitations. First, participants’ age was not included as a variable in the analyses. Mental health screening using the UPI and dental examinations were performed for freshmen at Yamagata University. Although most university freshmen are assumed to be 18, 19, or 20 years old, several of them may be older. The second limitation is the lack of data on dental examinations in 2015 and 2016 because of system failure. This reflects a potential risk that the present results do not represent all students. Third, the final analysis was performed only for 12,433 (65.9%) of 18,873 students. As described above, this means that the final sample may not be representative of the original target population, which included all students. The fourth limitation is measurement bias. In the present study, dental health examinations were performed by several dentists. The diagnostic criteria followed by all the dentists are not likely to have been a perfect match. However, students who displayed subjective symptoms, such as disturbance in opening the mouth, pain in opening the jaw, temporomandibular tenderness on self-palpation, gingival swelling and bleeding, and marked tartar deposition, were diagnosed and recommended for treatment. These criteria were very simple and easy to recognize; therefore, a measurement bias might not have existed among the dentists. The fifth limitation is that we did not check to see whether the fatigue effect occurred for students answering the questionnaires. The questionnaires should take approximately 10 min to complete; thus, we did not anticipate the fatigue effect. However, the UPI is still relatively lengthy [3], and some students therefore might have felt fatigued and ticked the same answer to finish faster. Future studies using this kind of questionnaire should check to see whether the fatigue effect occurred. Furthermore, the epidemiological design of our cross-sectional study is the sixth limitation. To improve the quality of the study’s evidence, students should be questioned at a minimum of two time points; therefore, this cross-sectional study should be developed into a longitudinal study in the future.

## 5. Conclusions

This cross-sectional study revealed that students who required treatment for TMD had a risk of obtaining >30 UPI scores. As a whole, our hypothesis (students requiring treatment for dental health will have a higher risk of >30 UPI scores, which means they will be advised to consult mental health professionals, compared to those who do not require treatment) was appropriate; however, the evidence is limited because there were several limitations in this study. The symptoms (requiring treatment) of TMD, such as disturbance in opening the mouth, pain in opening the jaw, and temporomandibular tenderness on self-palpation, can be easily recognized by students themselves. Students who have a UPI score above 30 are recommended to consult a mental health professional. Considering the relationship between TMD and UPI scores in this study, it may be possible to perform pre-screening for mental health based on the subjective symptoms of TMD before screening using the UPI. Students who have some subjective symptoms of TMD may be strongly recommended to take the UPI test. Alternatively, they may be recommended to consult mental health professionals.

## Figures and Tables

**Table 1 ijerph-19-04622-t001:** Participants’ characteristics of dental health status stratified according to UPI scores.

		UPI Scores				
		<30		≥30		
Variable		n	%	n	%	*p*-Value ^†^
Sex	Male	6910	62.1	684	52.4	<0.001 *
	Female	4217	37.9	621	47.6	
	Not available	1	0.0	0	0.0	
Dental caries	No abnormality	8587	77.2	1000	76.6	0.902
	Required observation	1109	10.0	132	10.1	
	Required treatment	1432	12.9	173	13.3	
Periodontal disease	No abnormality	6969	62.6	789	60.5	0.305
	Required observation	2155	19.4	265	20.3	
	Required treatment	2004	18.0	251	19.2	
Malocclusion	No abnormality	9936	89.3	1173	89.9	0.375
	Required observation	1032	9.3	109	8.4	
	Required treatment	160	1.4	23	1.8	
Temporomandibular disorders	No abnormality	10,863	97.6	1262	96.7	<0.001 *
	Required observation	227	2.0	29	2.2	
	Required treatment	38	0.3	14	1.1	

^†^ *p*-value based on the chi-squared test. UPI, University Personality Inventory. * Statistically significant.

**Table 2 ijerph-19-04622-t002:** Participants’ characteristics of dental health status stratified according to sex.

		Sex						
		Male		Female		Not Available		
Variable		n	%	n	%			*p*-Value ^†^
Dental caries	No abnormality	5886	77.5	3701	76.5	0	0.0	<0.001 *
	Required observation	695	9.2	546	11.3	0	0.0	
	Required treatment	1013	13.3	591	12.2	1	100.0	
Periodontal disease	No abnormality	4392	57.8	3365	69.6	1	100.0	<0.001 *
	Required observation	1635	21.5	785	16.2	0	0.0	
	Required treatment	1567	20.6	688	14.2	0	0.0	
Malocclusion	No abnormality	6835	90.0	4273	88.3	1	100.0	0.052
	Required observation	650	8.6	491	10.1	0	0.0	
	Required treatment	109	1.4	74	1.5	0	0.0	
Temporomandibular disorders	No abnormality	7437	97.9	4687	96.9	1	100.0	0.004 *
	Required observation	135	1.8	121	2.5	0	0.0	
	Required treatment	22	0.3	30	0.6	0	0.0	

^†^ *p*-value based on the chi-squared test. * Statistically significant.

**Table 3 ijerph-19-04622-t003:** Crude and adjusted odds ratios and 95% confidence intervals of the variables associated with having >30 UPI scores.

Variable	Crude OR ^†^	95% CI	*p*-Value	Adjusted OR ^‡^	95% CI	*p*-Value
All participants						
Dental caries	1.305	(0.874–2.099)	0.692			
Periodontal disease	1.084	(0.937–1.255)	0.277			
Malocclusion	1.230	(0.791–1.911)	0.358			
Temporomandibular disorders	3.165	(1.710–5.857)	<0.001 *			
Male participants						
Dental caries	1.110	(0.887–1.389)	0.361			
Periodontal disease	1.328	(1.107–1.594)	0.002 *	1.329	(1.108–1.595)	0.002 *
Malocclusion	1.375	(0.766–2.467)	0.286			
Temporomandibular disorders	2.986	(1.098–8.118)	0.032 *	3.014	(1.107–8.204)	0.031 *
Female participants						
Dental caries	0.968	(0.747–1.255)	0.807			
Periodontal disease	0.864	(0.672–1.110)	0.252			
Malocclusion	1.062	(0.542–2.079)	0.861			
Temporomandibular disorders	2.938	(1.340–6.445)	0.007 *			

^†^ OR for “Required treatment” (vs. “Required observation”/”No abnormality”). ^‡^ Adjusted for variables with *p* < 0.05 in the univariate analysis. Adjusted ORs were not calculated in all and female participants because only one variable was significant in the univariate analysis. * Statistically significant. OR, odds ratio; CI, confidence interval.

## Data Availability

The raw data are confidential and cannot readily be shared. Researchers need to obtain permission from the Institutional Review Board and apply for access to the data from the Ethics Committee of Yamagata University.

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
