# Peer review of "A Cross-Sectional Survey on the Association between Dental Health Conditions and University Personality Inventory Scores among University Students: A Single-Center Study in Japan"

_ijerph, 2022, doi:10.3390/ijerph19084622_

Round 1

Reviewer 1 Report

Dear Authors,

Thank you for reviewing your manuscript. It was interesting to read, well written. I have only comments regarding methodology. First, in the methods section you write that there was one dentist who performed dental examination and in the discussion you write in plural form which suggest more than one dentist performing dental examination. If there were more than one dentist, were they calibrated, if not, why.

Second, can you write more detailed about how you detected dental caries and periodontal disease in patients. I consider this quite important for the reader

Third, the questionnaire was quite long in my opinion. Please add how long time it took to fill it and have you found fatigue effect? This is situation where respondents tend to tick the same answer to finish faster if they have to fill long questionnaires.

Thank you very much

Author Response

Reviewer 1

Dear Authors,

Thank you for reviewing your manuscript. It was interesting to read, well written. I have only comments regarding methodology. First, in the methods section you write that there was one dentist who performed dental examination and in the discussion you write in plural form which suggest more than one dentist performing dental examination. If there were more than one dentist, were they calibrated, if not, why.

Response:

Thank you for your helpful comment. We apologize for our misleading wording.

One dentist conducted the dental examination for one student. Multiple dentists did not conduct the dental examinations for one student. For many students, several dentists conducted the dental examinations.

Furthermore, before the dental examinations, the dentists met and confirmed the criteria for the diagnosis of dental caries, periodontal disease, malocclusion, and temporomandibular disorders. However, as we note in our discussion of the limitations in the Discussion section, there might be a measurement bias.

According to your comments, we revised the manuscript as follows.

Lines 79 to 80

Medical and dental check-ups were performed by doctors and dentists, respectively, for each student one month after enrollment.

Lines 89 to 91

To minimize the measurement bias, before the dental examinations, dentists met to confirm the diagnostic criteria for dental caries, periodontal disease, malocclusion, and TMD. 

Second, can you write more detailed about how you detected dental caries and periodontal disease in patients. I consider this quite important for the reader

Response:

Thank you for your helpful comment. We detected dental caries and periodontal disease by using a portable mobile dental examination light, dental mirror, dental explorer, and dental probe. According to your suggestions, we have added the following details.

Lines 91 to 94

To detect dental caries and periodontal disease, dentists used a portable mobile dental examination light, dental mirror, dental explorer, and dental probe. Dentists detected malocclusion and temporomandibular disorders by visual and tactile examination.

Third, the questionnaire was quite long in my opinion. Please add how long time it took to fill it and have you found fatigue effect? This is situation where respondents tend to tick the same answer to finish faster if they have to fill long questionnaires.

Response:

Thank you for your helpful comment. The questionnaire, UPI, is a 60-item self-report questionnaire that uses a binary response scale (yes or no). It takes approximately 10 minutes to complete. Therefore, we do not believe that completing the questionnaire was a huge burden for our students. However, because we did not confirm how long it took each student to fill it out and we did not check for the fatigue effect, we cannot definitively answer your question. As you mentioned, some students might have felt fatigued and ticked the same answer to finish faster. We think this is an important limitation.

According to your suggestion, we revised our discussion of limitations as follows.

Lines 226 to 232

The fifth limitation is that we did not check to see whether the fatigue effect occurred for students answering the questionnaires. The questionnaires should take approximately 10 minutes to complete; thus, we did not anticipate the fatigue effect. However, the UPI is still relatively lengthy [3], and some students therefore might have felt fatigued and ticked the same answer to finish faster. Future studies using this kind of questionnaire should check to see whether the fatigue effect occurred.

Thank you very much

We appreciate your kind and polite review.

Reviewer 2 Report

“The association between dental health conditions and University Personality Inventory scores among university students in Japan” was submitted to IJERPH

This study aimed to examine the association between dental health conditions and scores on the UPI in students of a Japanese university.

The manuscript deals with an interesting issue; however, there are several concerns related to the study.

Title: Please include the epidemiological design. Moreover, It is necessary to clarify that the study was carried out on students from only one university in Japan.

Abstract/Results: it is necessary to include the confidence intervals of each OR.

Abstract/Discussion: Clarify that the study has limitations

Keywords: University Personality Inventory; dental health condition, mental health screening; university students are not MeSH terms.

Introduction

The need to assess university students through UPI is not sufficiently supported

Methods

Why the different dentists were not calibrated in the diagnoses? Please explain the reasons (Major Concern). This aspect generates many biases in the study.

Were statistical tests performed to establish the distribution of the data?

Were the assumptions of the regression model checked? If so, indicate the process.

Results

It would be desirable to know the periodontal status stratified according to sex.

Discussion.

The most important results are discussed without a scientific basis. An additional limitation of the study is its epidemiological design.

Author Response

Reviewer 2

“The association between dental health conditions and University Personality Inventory scores among university students in Japan” was submitted to IJERPH

This study aimed to examine the association between dental health conditions and scores on the UPI in students of a Japanese university.

The manuscript deals with an interesting issue; however, there are several concerns related to the study.

Title: Please include the epidemiological design. Moreover, It is necessary to clarify that the study was carried out on students from only one university in Japan.

Response:

Thank you for your helpful comment. We have revised the title as follows.

Lines 2 to 4

A cross-sectional survey on the association between dental health conditions and University Personality Inventory scores among university students: A single-center study in Japan

Abstract/Results: it is necessary to include the confidence intervals of each OR.

Response:

Thank you for your helpful comment. We have added the confidence intervals of each OR. Please confirm the revised abstract.

Abstract/Discussion: Clarify that the study has limitations

Response:

Thank you for your helpful comments. According to your suggestions, we clarified that our present study had limitations as follows.

Lines 26 to 28

Although our study has several limitations, students with subjective symptoms (e.g., disturbance in opening the mouth) should take the UPI test or in some cases consult mental health professionals.

Keywords: University Personality Inventory; dental health condition, mental health screening; university students are not MeSH terms.

Response:

Thank you for your helpful comment. We have revised the keywords so that they are all MeSH terms as follows.

Lines 29 to 30

Keywords: surveys and questionnaires; oral health; temporomandibular joint disorders; mental health; students

Introduction

The need to assess university students through UPI is not sufficiently supported

Response:

Thank you for your helpful comment. We have revised the Introduction to detail this need as follows.

Lines 44 to 46

Therefore, it is important to pay attention to university students who may be dealing with stress and it is also crucial to validate mental health using simplified questionnaires, such as the University Personality Inventory (UPI) discussed below.

Methods

Why the different dentists were not calibrated in the diagnoses? Please explain the reasons (Major Concern). This aspect generates many biases in the study.

Response:

Thank you for your helpful comment. We apologize for the insufficient description of the calibration of the diagnosis for dental examinations. Before the dental examinations, the dentists met to confirm the criteria for the diagnoses of dental caries, periodontal disease, malocclusion, and temporomandibular disorders. However, because it was difficult to perfectly match these diagnoses, we clarify this difficulty in our discussion of the limitations of our study. We have revised the text as follows.

Lines 89 to 91

To minimize the measurement bias, before the dental examinations, dentists met to confirm the diagnostic criteria for dental caries, periodontal disease, malocclusion, and TMD.

Were statistical tests performed to establish the distribution of the data?

Response:

Thank you for your helpful comment. We confirmed the distribution of the data from the UPI by using the Kolmogorov-Smirnov test. The p-value of the test is <0.001. Therefore, the distribution of the data from the UPI was not normal. However, our statistical analysis method was a generalized linear model of regression, not ordinary linear regression. Therefore, we do not believe this is problematic. We have added descriptions of these processes in the Materials and Methods section as follows.

Line 109

The Kolmogorov-Smirnov test was performed to confirm the distribution.

Lines 123 to 124

The p-value of the Kolmogorov-Smirnov test was <0.001; therefore, the distribution of data from the UPI was not normal.

Were the assumptions of the regression model checked? If so, indicate the process.

Response:

Thank you for your helpful comment. The logistic regression requires there to be little or no multicollinearity among the independent variables. Therefore, we confirmed that the values of variance inflation factors (VIF) of all independent variables were less than 10. Furthermore, we confirmed the interaction between the variables in the multivariate logistic regression analysis. The multivariate logistic regression analysis was performed for the male participants’ analysis (please see Table 2 [Male participants]); therefore, a two-way analysis of variance was performed to confirm the interaction between periodontal disease and temporomandibular disorders. In that analysis, the F- and p-values were 0.125 and 0.723, respectively. Therefore, we judged that there was no interaction between the variables. We added the above process to the Materials and Methods and Results sections as follows.

Lines 116 to 119

The logistic regression requires there to be little or no multicollinearity among the independent variables. Therefore, we confirmed that the values of variance inflation factors of all independent variables were less than 10. Furthermore, the interaction between the variables in the multivariate logistic regression analysis was tested using a two-way analysis of variance.  

Lines 136 to 138

The interaction effect between [Periodontal disease] and [Temporomandibular disorders] in male participants was not confirmed (P=0.723).

Results

It would be desirable to know the periodontal status stratified according to sex.

Response:

Thank you for your helpful comment. We revised Table1b to show the dental health status stratified according to sex.

We have also added the following to the Results section.

Lines 126 to 129

Table 1b shows the characteristics of dental health status stratified according to sex. Although the statistical significance was confirmed from the presence of [Not available] on sex, the distribution of dental health status between [Male] and [Female] was very similar.

Discussion.

The most important results are discussed without a scientific basis. An additional limitation of the study is its epidemiological design.

Response:

Thank you for your helpful comments. Our most important results are that students who required treatment for TMD had an approximately three-fold higher risk of >30 UPI scores than those who had no abnormality or required observation. We added a discussion about the association between TMD and mental health (with a scientific basis) as follows. Further, we added a discussion about the limitation of the epidemiological design as follows.

Lines 170 to 189

Correlations between dental health conditions and mental health have been reported in several studies [13-17]. In particular, the association between TMD and mental health has been widely reported [20-22], although the precise mechanisms between mental health and TMD remain unclear [21]. The etiology of TMD is multifactorial and is well-known to comprise not only biological factors, such as derangements in the temporomandibular joint, but also psychological factors, such as depression, anxiety, and stress [21]. There is a clear link between stress and bruxism, which may explain TMD. Stress has the potential to alter the threshold of pain perception in the central nervous system and increases the intensity of parafunctional habits such as bruxism [21]. Colonna et al. found that almost 50% of the 506 participants with poor mental health due to the COVID-19 pandemic reported an increase in bruxism behaviors, and approximately 30% of the participants reported an increase in their TMD symptoms [23]. Carter also reported an association between stress and bruxism among veterans with Gulf War Illness [24]; the participants, who had higher levels of perceived stress than the general population, demonstrated a high frequency of teeth grinding and clenching. Bruxism puts pressure on the orofacial muscle, and persistent pressure on this muscle leads to pain in the masseter and temporal muscles. Furthermore, persistent pressure on the mandible also leads to temporomandibular joint pain. These pains are some of the typical symptoms of TMD. Considering these past reports, the results of the present study regarding the association between TMD and UPI can be considered reasonable and logical. 

Lines 232 to 235

Furthermore, the epidemiological design of our cross-sectional study is the sixth limitation. To improve the quality of the study’s evidence, students should be questioned at a minimum of two time points; therefore, this cross-sectional study should be developed into a longitudinal study in the future.

Reviewer 3 Report

First of all, thank you for the opportunity to review this paper.

The purpose of this cross-sectional study was to determine the associations between dental health conditions and the University Personality Inventory (UPI) scores for first-year students at Yamagata University, Japan.

The work is interesting but not well designed. It has been conducted for many years, and probably some parts cannot be repeated. It is a pity that students were not questioned at a minimum of two-time points to draw better conclusions. However, given the data the authors have, the paper is solidly written.

Introduction:

  1. Line 57 - 63 - belong more to the methodology than to the introduction - please correct the same.
  2. Did this paper have a hypothesis, if so, which one?

Methods:

  1. Please indicate which period an examination was conducted for all freshmen at the beginning of the academic year or the end of the first academic year.
  2. Why didn’t you use WHO-recommended indexes like DMFT?
  3. Were there any underage respondents? Did respondents sign informed consent?
  4. Why were only freshmen questioned? Why was the examination not repeated in one of the following years of study, perhaps the last one?

Discussion:

  1. In the discussion, elaborate the obtained results with other studies in more detail. I also ask you to confirm the hypothesis of the study.

References:

  1. Please write references following the instructions of the journal. Behind the last author's name is a period, not a comma.

Author Response

Reviewer 3

First of all, thank you for the opportunity to review this paper.

The purpose of this cross-sectional study was to determine the associations between dental health conditions and the University Personality Inventory (UPI) scores for first-year students at Yamagata University, Japan.

The work is interesting but not well designed. It has been conducted for many years, and probably some parts cannot be repeated. It is a pity that students were not questioned at a minimum of two-time points to draw better conclusions. However, given the data the authors have, the paper is solidly written.

Introduction:

  1. Line 57 - 63 - belong more to the methodology than to the introduction - please correct the same.

Response:

Thank you for your helpful comment. As you mentioned, this description belongs in the methodology section. We have moved it to the Materials and Methods section as follows.

Lines 75 to 79

At Yamagata University, medical and dental check-ups are performed for all freshmen every year. Furthermore, mental health screening using the UPI is also performed for all freshmen every year; students with scores greater than 30 are recommended to mental health professionals. We used these data on dental examination and UPI for the present study.

  1. Did this paper have a hypothesis, if so, which one?

Response:

Thank you for your helpful comments. We did not clarify our hypothesis. According to your suggestion, we have added our hypothesis in the Introduction section as follows.

Lines 58 to 62

Therefore, we hypothesized the following: students requiring treatment for dental health will have a higher risk of >30 UPI scores, which means they will be advised to consult mental health professionals, compared to those who did not require treatment. If a UPI score above 30 is related to subjective dental symptoms, students with such symptoms may be considered at risk of poor mental health.

Methods:

  1. Please indicate which period an examination was conducted for all freshmen at the beginning of the academic year or the end of the first academic year.

Response:

Thank you for your helpful comment. All freshmen were given an examination one month after enrollment in the university. We had described this information in the Materials and Methods section (Page, line 78 to line 80).

  1. Why didn’t you use WHO-recommended indexes like DMFT?

Response:

Thank you for your helpful query. Because we had a limited time to conduct our study and only two or three dentists were available to examine approximately four or five hundred students, simple screening was conducted conventionally. The original aim was to screen for dental caries; therefore, it is sufficient to determine whether dental caries are present. However, to conduct a more detailed survey, WHO-recommended indexes, such as DMFT, should be adopted in the future, as you mentioned.

  1. Were there any underage respondents? Did respondents sign informed consent?

Response:

Thank you for your query. As we detail in the Discussion section, we have no data on age. Therefore, we do not know the exact age distribution. However, most university freshmen in Japan are assumed to be 18–20 years old. Therefore, there were assumed to be many underage respondents. In this study, as we have already described, all students were given the option to opt-out of this study online, and no participants declined to participate. Informed consent was not signed for all students in this study.

  1. Why were only freshmen questioned? Why was the examination not repeated in one of the following years of study, perhaps the last one?

Response:

Thank you for your helpful query. It would be preferable to repeat the examination in one of the following years of study. However, because we used historical data on examinations for the freshmen in our university in this study, longitudinal examination was impossible. Nevertheless, your suggestion is very important for improving research quality. Therefore, we will repeat the examination in one of the following years of study in the future. Furthermore, your suggestion highlights an important limitation of our research; therefore, we have added this limitation in our Discussion section as follows.

Lines 232 to 235

Furthermore, the epidemiological design of our cross-sectional study is the sixth limitation. To improve the quality of the study’s evidence, students should be questioned at a minimum of two time points; therefore, this cross-sectional study should be developed into a longitudinal study in the future.

Discussion:

  1. In the discussion, elaborate the obtained results with other studies in more detail. I also ask you to confirm the hypothesis of the study.

Response:

Thank you for your helpful comments. According to your suggestions, we elaborated on our results using other studies as follows. Furthermore, we also clarified our hypothesis in the Introduction section and confirmed it as follows.

Lines 170 to 189

Correlations between dental health conditions and mental health have been reported in several studies [13-17]. In particular, the association between TMD and mental health has been widely reported [20-22], although the precise mechanisms between mental health and TMD remain unclear [21]. The etiology of TMD is multifactorial and is well-known to comprise not only biological factors, such as derangements in the temporomandibular joint, but also psychological factors, such as depression, anxiety, and stress [21]. There is a clear link between stress and bruxism, which may explain TMD. Stress has the potential to alter the threshold of pain perception in the central nervous system and increases the intensity of parafunctional habits such as bruxism [21]. Colonna et al. found that almost 50% of the 506 participants with poor mental health due to the COVID-19 pandemic reported an increase in bruxism behaviors, and approximately 30% of the participants reported an increase in their TMD symptoms [23]. Carter also reported an association between stress and bruxism among veterans with Gulf War Illness [24]; the participants, who had higher levels of perceived stress than the general population, demonstrated a high frequency of teeth grinding and clenching. Bruxism puts pressure on the orofacial muscle, and persistent pressure on this muscle leads to pain in the masseter and temporal muscles. Furthermore, persistent pressure on the mandible also leads to temporomandibular joint pain. These pains are some of the typical symptoms of TMD. Considering these past reports, the results of the present study regarding the association between TMD and UPI can be considered reasonable and logical. 

Lines 58 to 62

Therefore, we hypothesized the following: students requiring treatment for dental health will have a higher risk of >30 UPI scores, which means they will be advised to consult mental health professionals, compared to those who did not require treatment. If a UPI score above 30 is related to subjective dental symptoms, students with such symptoms may be considered at risk of poor mental health.

Lines 239 to 243

As a whole, our hypothesis (students requiring treatment for dental health will have a higher risk of >30 UPI scores, which means they will be advised to consult mental health professionals, compared to those who do not require treatment) was appropriate; however, the evidence is limited because there were several limitations in this study.

References:

  1. Please write references following the instructions of the journal. Behind the last author's name is a period, not a comma.

Response:

Thank you for your helpful comments. We have revised the references as per the journal’s guidelines.

Round 2

Reviewer 2 Report

None